# Beat LLMs at Their Own Game: Zero-Shot LLM-Generated Text Detection via Querying ChatGPT

**Biru Zhu**[1], **Lifan Yuan**[2*] **Ganqu Cui**[2], **Yangyi Chen**[3], **Chong Fu**[4], **Bingxiang He**[2],
**Yangdong Deng**[1†] **Zhiyuan Liu**[2†], **Maosong Sun**[2], **Ming Gu**[1]

[1] School of Software, Tsinghua University, China
[2] Department of Computer Science and Technology, Tsinghua University, China
[3] University of Illinois Urbana-Champaign, USA [4] Zhejiang University, China
zbr19@mails.tsinghua.edu.cn, lievanyuan173@gmail.com
{dengyd, liuzy}@tsinghua.edu.cn

## Abstract

Large language models (LLMs), e.g., ChatGPT, have revolutionized the domain of natural language processing because of their excellent performance on various tasks. Despite their great potential, LLMs also incur serious concerns as they are likely to be misused. There are already reported cases of academic cheating by using LLMs. Thus, it is a pressing problem to identify LLM-generated texts. In this work, we design a zero-shot black-box method for detecting LLM-generated texts. The key idea is to revise the text to be detected using the ChatGPT model. Our method is based on the intuition that the ChatGPT model will make fewer revisions to LLM-generated texts than it does to human-written texts, because the texts generated by LLMs are more in accord with the generation logic and statistical patterns learned by LLMs like ChatGPT. Thus, if the text to be detected and its ChatGPT-revised version have a higher degree of similarity, the text is more likely to be LLM-generated. Extensive experiments on various datasets and tasks show that our method can effectively detect LLM-generated texts. Moreover, compared with other detection methods, our method has better generalization ability and is more stable across various datasets. The codes are publicly available at `https://github.com/thunlp/LLM-gene rated-text-detection`.

## 1 Introduction

Benefiting from the emergent ability, large language models (LLMs) have demonstrated excellent performance on a large number of natural language processing tasks (Brown et al., 2020; Sanh et al., 2022). Recently, LLMs (e.g., ChatGPT) trained to follow instructions (Ouyang et al., 2022; Wei et al., 2022) can generate high-quality responses when given specific instructions and input data by a user.

The strong emergent abilities, however, also trigger social concerns for the misuse of LLMs. The academic cheating by using LLMs to write essays and homework has been repeatedly reported. The fabricated news generated by LLMs also causes a significant problem to our society. Thus, it is an important problem to detect whether a piece of text is generated by the LLM. Following previous works (Jawahar et al., 2020; Mitchell et al., 2023), we formulate the problem of detecting LLM-generated texts as a binary classification task, i.e., classifying whether a piece of text is generated by the LLM or written by human.

Previous methods for detecting LLM-generated texts can be classified into two categories. The first category is the zero-shot detection method (Solaiman et al., 2019; Gehrmann et al., 2019; Mitchell et al., 2023), which needs to access the model's output logits or losses for detection. However, many LLM services provided by commercial companies do not expose the model's output logits or losses at the inference time. Thus, these methods have to rely on local proxy models to access the output information. However, the inconsistency between the online model and the local proxy model may lead to poor detection performance. The second category is the supervised fine-tuning method (Guo et al., 2023), which trains a Deep Neural Network (DNN)-based classifier with the labeled training data. It is prone to overfitting the training data and thus may have poor generalization ability.

To address the abovementioned problems, we propose a zero-shot black-box detection method leveraging the ChatGPT model. Our method is based on the following finding. If we use the ChatGPT model to revise texts, it will make fewer revisions to LLM-generated texts than it does to human-written texts. The underlying reason may be that the LLM-generated texts conform to the generation logic and statistical patterns learned by the ChatGPT model. The overall framework of our

---

*Lifan Yuan has graduated from Huazhong University of Science and Technology and he is doing an internship in THUNLP group now.
†Corresponding author.

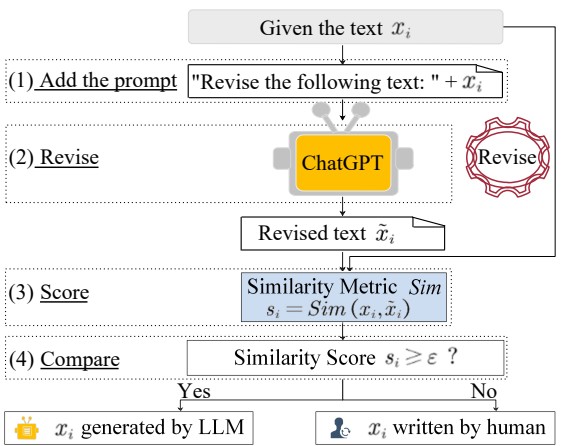

Figure 1: The overall framework of our approach. Firstly, the text to be detected $x_i$ is revised by ChatGPT. Then our method calculates the similarity score $s_i$ between the text $x_i$ and its corresponding revised text $\tilde{x}_i$. If the similarity score $s_i$ is no smaller than the threshold $\varepsilon$, the text $x_i$ is likely to be generated by LLM.

method is shown in Figure 1. Firstly, we leverage the ChatGPT model to revise the text to be detected. Then we measure the similarity between the text to be detected and its ChatGPT-revised version using the similarity metric. The applied similarity metric could be BLEU score (Papineni et al., 2002), ROUGE score (Lin, 2004), BERTScore (Zhang* et al., 2020) or BARTScore (Yuan et al., 2021). The LLM-generated texts are more likely to achieve higher similarity scores compared with human-written texts.

We perform experiments on six datasets including question answering and summarization generation tasks. The experimental results prove that our method can effectively detect LLM-generated texts. Besides, we compare our method with other zero-shot detection methods and the supervised fine-tuning method. The experimental results show that our detection method is more universal across various datasets and has better generalization ability. Moreover, our method is robust to dataset biases.

## 2 Related Work

**Large Language Models.** Large language models (LLMs) have achieved fantastic performance on a huge number of natural language processing tasks (Brown et al., 2020). Some recent works show that LLMs can be trained to follow instructions with instruction-tuning (Ouyang et al., 2022; Wei et al., 2022; Chung et al., 2022). LLMs provided by AI companies like OpenAI can generate responses to users' inputs and solve different prob-

lems. Many applications are built upon them, e.g., chatbots. Though LLMs act as useful tools, misusing them may cause bad influence, e.g., students use LLMs to do their homework.

**LLM-Generated Text Detection.** Previous LLM-generated text detection methods can generally be classified into two categories. The first category is the zero-shot detection method (Solaiman et al., 2019; Gehrmann et al., 2019; Mitchell et al., 2023). However, these methods rely on the model's output logits or losses. If there is no access to the output logits or losses of the source model that generates the text to be detected, these methods can use a proxy model to get the proxy model's output logits or losses for detection. However, the difference between the source model and the proxy model may cause poor detection performance. Thus, it is essential to design a detection method without accessing the model's output logits or losses.

The second category is to train a DNN-based classifier using some labeled human-written and LLM-generated samples (Guo et al., 2023; Uchendu et al., 2020). However, it is hard to collect sufficient labeled training samples to train a generalizable DNN-based classifier (Bakhtin et al., 2019; Uchendu et al., 2020). Moreover, training DNN-based classifiers are vulnerable to backdoor attacks (Qi et al., 2021) and the DNN-based detector is lack of interpretability. Thus, it is important to build an explainable detector without training DNN-based classifiers.

Jawahar et al. (2020) survey some automatic detection methods and identify some future directions for building useful detectors. He et al. (2023) develop a benchmark for evaluating existing detection methods and call for more robust detection methods. In this paper, we aim to design a robust and explainable black-box detection method without training DNN-based classifiers.

## 3 Method

In this section, we illustrate our method step by step. (1) Given the texts $\{x_1, x_2, ..., x_n\}$ to be detected, we first revise the texts using the ChatGPT model. Specifically, we add the prompt, i.e., "Revise the following text: ", before the texts to be detected. Then we feed the modified texts into the ChatGPT model $\mathcal{M}$ and use its responses $\{\tilde{x}_1, \tilde{x}_2, ..., \tilde{x}_n\}$ as the revised texts.

$$\tilde{x}_i = \mathcal{M}(\text{``Revise the following text: ''} + x_i),$$
$$i \in \{1, 2, ...n\} \tag{1}$$

(2) Then we use an unsupervised similarity metric $Sim$ to calculate the similarity score $s_i$ between the original text $x_i$ and its revised text $\tilde{x}_i$.

$$s_i = Sim\,(x_i, \tilde{x}_i) \tag{2}$$

We take the BARTScore as an example. Given a sequence-to-sequence pre-trained model like BART model (Lewis et al., 2020) which is parameterized by $\theta$, the BARTScore is calculated using the log probability of the target text $x_i$ given the revised text $\tilde{x}_i$ as the source text. The target text is tokenized into a sequence of tokens: $x_i = \{x_{i1}, x_{i2}, ..., x_{ik}\}$.

$$BARTScore = \sum_{t=1}^{k} \log p\left(x_{it} | x_{i(j<t)}, \tilde{x}_i, \theta\right) \tag{3}$$

The BARTScore can measure the semantic coverage between the source text and the target text (Yuan et al., 2021). For the calculation of BARTScore in this paper, we use the BARTScore-CNN, which uses a BART model that is fine-tuned on the CNNDM dataset (Hermann et al., 2015). For more details of other evaluated similarity metrics, please refer to appendix B. The LLM-generated text and its revised text are more similar compared with the human-written text and its revised text. Thus, a higher similarity score $s_i$ indicates that the text $x_i$ is more likely to be LLM-generated.

## 4 Experiments

In this section, we evaluate the detection performance of our method and compare our method with other detection methods.

### 4.1 Experimental Setting

**Datasets.** For the summarization generation task, we perform experiments on MultiNews (Fabbri et al., 2019), GovReport (Huang et al., 2021) and BillSum (Kornilova and Eidelman, 2019) datasets. We use the representative LLM, i.e., ChatGPT (gpt-3.5-turbo), as the source model to generate summaries to be detected. For the question answering task, we consider three datasets including Finance (Maia et al., 2018), Medicine (Chen et al., 2020), and Reddit Eli5 (Fan et al., 2019). In main experiments, we use the ChatGPT-generated texts

collected by Guo et al. (2023). For more details of the datasets, please refer to appendix D. We also demonstrate that our method can detect texts that are generated by other source models in section 4.3.

### 4.2 Main Experiments

#### 4.2.1 Comparisons in Zero-Shot Setting

**Zero-Shot Methods for Comparison.** We compare our method with other zero-shot detection methods including Log-Likelihood (Solaiman et al., 2019), Rank (Gehrmann et al., 2019), Log-Rank (Mitchell et al., 2023), Entropy (Gehrmann et al., 2019) and DetectGPT (Mitchell et al., 2023). These zero-shot detection methods are based on differences between output losses/logits of the model on human-written texts and LLM-generated texts. They consider that the model will be more familiar with LLM-generated texts. For example, the Log-Likelihood method takes the negative loss of the model on the text to be detected as the Log-Likelihood score. A higher Log-Likelihood score indicates the text is more likely to be LLM-generated. The details of these detection methods are shown in appendix C. Since the logits or losses of ChatGPT can not be accessed at the inference time, we use GPT-2-medium (Radford et al., 2019) as the proxy model for deriving the logits or losses for these methods, following He et al. (2023).

**Metrics.** Following Mitchell et al. (2023), we use the AUROC as the evaluation metric. For our method, we use the BARTScore-CNN (Yuan et al., 2021) as the similarity metric. We also prove that our method is effective under different similarity metrics. For more details of our method's detection performance when using different similarity metrics including BLEU, ROUGE and BERTScore, please refer to appendix A.

**Results.** As shown in Table 1, our method achieves good detection performance across various datasets. Specifically, the AUROC is consistently higher than 70% for our method on all datasets. The average AUROC on all datasets is 90.05% for our method. However, for other zero-shot detection methods, the AUROC is low on MultiNews, GovReport and BillSum datasets. The reason may be that the training data of ChatGPT is different from that of GPT-2-medium. Thus, the GPT-2-medium model may be unfamiliar with some ChatGPT-generated texts, reflected in the high losses on some ChatGPT-generated texts. As a result, the detec-

| Method | Finance | Medicine | Reddit Eli5 | MultiNews | GovReport | BillSum | Average Value |
|---|---|---|---|---|---|---|---|
| Log-Likelihood | 98.61 | 98.85 | 99.08 | 55.07 | 36.17 | 34.15 | 70.32 |
| Rank | 92.24 | 97.62 | 78.81 | 56.61 | 42.44 | 37.03 | 67.46 |
| Log-Rank | 98.64 | 98.91 | 99.26 | 56.12 | 36.93 | 35.79 | 70.94 |
| Entropy | 97.07 | 98.79 | 97.78 | 52.31 | 31.54 | 24.50 | 67.0 |
| DetectGPT | 88.56 | 96.43 | 83.66 | 40.36 | 43.63 | 36.57 | 64.87 |
| Our Method (BARTScore-CNN) | 97.40 | 95.06 | 99.15 | 86.28 | 75.64 | 86.77 | **90.05** |

Table 1: Comparisons with other zero-shot detection methods. The evaluation metric is AUROC (%). The "Average Value" is the average performance on all datasets.

| Source Dataset | Fine-Tuning | Our Method |
|---|---|---|
| Finance | 81.77 | 80.05 |
| Medicine | 67.35 | 80.62 |
| Reddit Eli5 | 52.68 | 69.93 |
| MultiNews | 51.21 | 83.44 |
| GovReport | 83.62 | 82.67 |
| BillSum | 65.03 | 81.39 |

Table 2: The average accuracy (%) on all target datasets of the fine-tuning method and our method in the OOD setting when using original datasets as source datasets.

| Source Dataset | Fine-Tuning | Our Method |
|---|---|---|
| Biased Finance | 54.89 | 80.36 |
| Biased Medicine | 56.31 | 80.48 |
| Biased GovReport | 54.34 | 85.74 |

Table 3: The average accuracy (%) on all target datasets of the fine-tuning method and our method in the OOD setting when using biased datasets as source datasets.

tion performance of the methods that rely on the output losses of the proxy model will degrade. For the statistics of BARTScores and Log-Likelihood scores, please refer to appendix A.

### 4.2.2 Comparisons with Fine-Tuning

**Comparisons in Vanilla Setting.** We compare the accuracy of our method with the fine-tuning method. To calculate the accuracy of our method, we need to determine a threshold. If the test sample's similarity score between it and its revised text is no smaller than the threshold, it is predicted as the LLM-generated text. The detailed steps to find the optimal threshold on the source dataset are as follows[1]. Firstly, we get the similarity scores for all samples in the source dataset and their ChatGPT-revised versions. We take the minimum similarity score and the maximum similarity score. Then, we split the range between the minimum similar-

ity score and the maximum similarity score into 10,000 uniform intervals. We get 10,000 interpolation values between the minimum similarity score and the maximum similarity score. Finally, we use the best interpolation value with the highest classification accuracy on the source dataset as the optimal threshold.

We find that the fine-tuning method performs well in the situation where the distributions of the training and testing data are identical, as shown in appendix A. However, in practice, it is difficult to guarantee that the distributions of the training and testing data are identical. Thus, we also test the out-of-distribution (OOD) robustness. Specifically, we use one dataset as the training (source) dataset and the other datasets as the testing (target) datasets.

As shown in Table 2, the OOD performance of our method is better than the fine-tuning method on Medicine, Reddit Eli5, MultiNews and BillSum datasets. The fine-tuning method may overfit the training datasets and thus has poor OOD performance. Overall, our method is more robust under the OOD setting. The reason is that the difference between degrees of revisions for the human-written texts and LLM-generated texts is the universal feature among various datasets.

**Comparisons on Biased Datasets.** The fine-tuning method is proven to easily overfit the dataset biases (McCoy et al., 2019). The dataset biases are some spurious correlations that are not shared among all datasets of a task (Lynch et al., 2023). We construct the biased datasets by adding the prefix "Answer: " to the human-written answers of Finance and Medicine datasets and adding the prefix "Summarization: " to the human-written summaries of the GovReport dataset.

As shown in Table 3, for the fine-tuning method, the classification models that are fine-tuned on biased datasets have very poor OOD performance, with all average accuracy falling below 60%. The reason is that the trained classification model with

---

[1]For the fine-tuning method, the source dataset is the training dataset, which is used to train a classifier. Our method does not need to train a classifier. For our method, we find the optimal threshold on the source dataset.

| Source Model | BERTScore | | BARTScore-CNN | |
| --- | --- | --- | --- | --- |
| | Finance | Medicine | Finance | Medicine |
| Text-davinci-003 | 90.18 | 92.94 | 83.38 | 91.41 |
| Text-davinci-002 | 86.83 | 86.21 | 88.90 | 90.81 |
| Vicuna | 78.68 | 88.40 | 95.43 | 97.88 |

Table 4: Performance of our method when detecting texts that are generated by various source models. The evaluation metric is AUROC (%).

the fine-tuning method overfits dataset biases, i.e., the mapping from the prefix to the label "human-written". However, these dataset biases are not universal features among all datasets. Also, from the results in Table 3, we can see that our method is robust to the dataset bias. The average accuracy on all target datasets of our method is above 80% with whichever biased dataset as the source dataset. The reason is that our method leverages the similarity between a pair of texts and is less influenced by the inserted prefix biases. For more details, please refer to appendix A.

### 4.3 Detecting Texts Generated by Various Source Models

We demonstrate that our method can be applied to detect texts generated by various source models such as text-davinci-003, text-davinci-002 and Vicuna (Chiang et al., 2023). Firstly, we use the source model to generate answers of the Finance and Medicine datasets. Then we leverage ChatGPT to revise texts. As shown in Table 4, our method can detect texts generated by various source models. The AUROC is higher than 78% under all the evaluated settings. The revisions for other source models' generated texts are minor compared with the revisions for human-written texts when using ChatGPT to revise the texts. The texts generated by ChatGPT and other source models may share some similar characteristics. The deeper reason may be that the training data of ChatGPT and other source models may have similar or overlapping parts.

### 4.4 Case Study

We conduct the case study on a human-written summary and the summary generated by Chat-GPT for the same bill from the BillSum dataset. From Figure 3 in the appendix, we can see that for the ChatGPT-generated summary, the revisions are mostly about synonym replacements or adding a word. However, from Figure 2 in the appendix, we can see that the revision is relatively large for the

human-written summary, which may change the sentences' structures. For example, a long human-written sentence with complex logic is revised into a relatively shorter sentence by ChatGPT. For more details, please refer to appendix A.

## 5 Conclusion

In this paper, we design a simple and effective baseline method for detecting LLM-generated texts. Our proposed method is based on the intuition that the ChatGPT model revises less for LLM-generated texts than it does for human-written texts. Compared with previous methods, our method neither needs to train a DNN-based classifier nor requires access to the source model's output logits or losses. Thus, our method has better generalization ability and is more practical. We perform experiments on six datasets of question answering and summarization generation tasks. The experimental results on various datasets show that our method can effectively detect LLM-generated texts. Moreover, our method is robust to dataset biases.

## 6 Limitations

Although our method is simple and effective, it still has some limitations. Firstly, our method needs to query the ChatGPT model and thus may cost money. Secondly, if the text to be detected is extremely short, the difference of revisions for human-written and LLM-generated texts may not be evident, which may lower the detection performance of our method.

## Acknowledgments

This work is supported by the National Key R&D Program of China (No.2022ZD0116312), National Natural Science Foundation of China (No. 62236004) and Institute Guo Qiang at Tsinghua University.

Biru Zhu, Lifan Yuan and Chong Fu designed the methods. Biru Zhu, Lifan Yuan, Ganqu Cui, Yangyi Chen and Bingxiang He designed the experiments. Yangdong Deng, Zhiyuan Liu, Maosong Sun and Ming Gu advised the project and participated in the discussion.

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

# Appendix

## A  Additional Experimental Results

**Results of Different Similarity Metrics.** We evaluate the effectiveness of our method with different similarity metrics, including BLEU score, ROUGE score, BERTScore and BARTScore-CNN. We use the similarity score as the prediction score and the labels of LLM-generated texts are set as 1. The experimental results are shown in Table 5. From the experimental results, we can see that the AUROC is above 60% no matter which similarity metric is used on all datasets, which demonstrates the effectiveness of our method. The results verify our intuition, i.e., compared with human-written texts and their corresponding revised texts, LLM-generated texts and their corresponding revised texts are more similar.

Besides, we can find that the performance of our detection method varies with different similarity metrics. For example, when the similarity metric is BARTScore-CNN, the average AUROC on all datasets is 90.05%. When the similarity metric is BLEU score, the average AUROC on all datasets is 77.29%. The BLEU score and ROUGE score are based on n-gram matches. The BERTScore and BARTScore-CNN are DNN-based similarity metrics. To sum up, the DNN-based similarity metrics perform better than those based on n-gram matches. The reason may be that the DNN-based similarity metrics can capture the semantic similarity better than those based on n-gram matches. The BERTScore performs better than BARTScore-CNN on MultiNews, GovReport and BillSum datasets. However, the BARTScore-CNN achieves the highest average performance among all similarity metrics.

**Statistics of BARTScores.** For both human-written texts and ChatGPT-generated texts, we record the mean value and variance of BARTScores between the original texts and their corresponding revised texts. We use BARTScore-CNN as the similarity metric to calculate the BARTScores. The higher BARTScore indicates the higher similarity between the text and its corresponding revised text. As shown in Table 6, the average BARTScore of ChatGPT-generated texts and their revised texts is higher than that of human-written texts for each dataset. The ChatGPT model revises more for human-written texts compared with ChatGPT-generated texts. Also, the variance of BARTScores

of ChatGPT-generated texts and their revised texts is lower than that of human-written texts for each dataset. This means that the extents of revisions vary largely among human-written texts while varying little among ChatGPT-generated texts.

**Log-Likelihood Scores on Different Datasets.** The Log-Likelihood method takes the negative loss of the model on the text to be detected as the Log-Likelihood score. A higher Log-Likelihood score means this piece of text is more likely to be LLM-generated. Since the output losses of ChatGPT are not available at the inference time, we use the GPT-2-medium model as the proxy model for the Log-Likelihood method. We record the mean value of negative losses of the GPT-2-medium model (i.e., the Log-Likelihood scores) on human-written texts and that on ChatGPT-generated texts. As shown in Table 7, the losses derived from the GPT-2-medium model are high on the ChatGPT-generated texts of MultiNews, GovReport and BillSum.

**Results of Comparisons with Fine-Tuning in Vanilla Setting.** We compare the accuracy (ACC) of our method with the fine-tuning method. Besides considering the setting that the training (source) and testing (target) data have the same distribution, we also test OOD robustness. For the experiments where the training (source) and testing (target) datasets are of the same distribution, we evaluate the detection performance in the classic cross-validation manner. For the experiments of testing OOD robustness, we use one dataset as the training dataset (source dataset) and a different dataset as the testing dataset (target dataset).

The experimental results are shown in Table 8. We can see that the fine-tuning method achieves good performance when the training and testing data have the same distribution. However, as shown in Table 8, there are 14 results whose accuracy are below 60% for the fine-tuning method. Also, we can see that there are 2 results whose accuracy are below 60% for our method. From this point of view, our method has better generalization ability compared with the fine-tuning method. The trained classification model is easy to overfit the training data for the fine-tuning method. Thus, the generalization ability of the fine-tuning method is poor. Our method is more robust under the OOD setting.

**Results of Comparisons with Fine-Tuning on Biased Datasets.** The dataset biases are some spurious correlations that are not shared among

| Similarity Metric | Finance | Medicine | Reddit Eli5 | MultiNews | GovReport | BillSum | Average Value |
|---|---|---|---|---|---|---|---|
| BLEU Score | 83.93 | 90.18 | 83.80 | 70.73 | 69.55 | 65.54 | 77.29 |
| ROUGE-1 F1 Score | 85.43 | 90.28 | 82.34 | 73.16 | 68.69 | 70.25 | 78.36 |
| ROUGE-2 F1 Score | 87.60 | 90.18 | 81.17 | 69.85 | 65.18 | 63.40 | 76.23 |
| ROUGE-L F1 Score | 87.23 | 87.48 | 80.43 | 72.10 | 64.72 | 66.13 | 76.35 |
| BERTScore | 84.41 | 89.47 | 89.87 | **86.73** | **83.55** | **91.62** | 87.61 |
| BARTScore-CNN | **97.40** | **95.06** | **99.15** | 86.28 | 75.64 | 86.77 | **90.05** |

Table 5: Performance of our method using different similarity metrics. The evaluation metric is AUROC (%). The "Average Value" is the average performance on all datasets.

| Dataset | H-Mean | H-Var | C-Mean | C-Var |
|---|---|---|---|---|
| Finance | -2.504 | 0.271 | -1.373 | 0.088 |
| Medicine | -2.780 | 0.493 | -1.630 | 0.339 |
| Reddit Eli5 | -3.373 | 0.331 | -1.651 | 0.161 |
| MultiNews | -2.250 | 0.209 | -1.614 | 0.127 |
| GovReport | -1.826 | 0.218 | -1.411 | 0.125 |
| BillSum | -2.004 | 0.200 | -1.401 | 0.099 |

Table 6: Statistics of BARTScores computed on human-written/ChatGPT-generated texts and their corresponding revised texts. The prefix "H" and "C" represent "human-written" and "ChatGPT-generated", respectively. "Mean" and "Var" represent mean value and variance, respectively.

| Dataset | H-Score | C-Score |
|---|---|---|
| Finance | -3.355 | -1.864 |
| Medicine | -3.888 | -1.893 |
| Reddit Eli5 | -3.689 | -2.062 |
| MultiNews | -2.994 | -2.941 |
| GovReport | -2.634 | -2.788 |
| BillSum | -2.526 | -2.694 |

Table 7: The mean value of negative losses (Log-Likelihood scores) on human-written texts and that on ChatGPT-generated texts. "H-Score"/"C-Score" represents the mean value of negative losses on human-written texts/ChatGPT-generated texts.

all datasets of a specific task (McCoy et al., 2019; Lynch et al., 2023). We select 3 datasets as the source datasets that the classification model is fine-tuned on, including Finance, Medicine, and Gov-Report. The classification models trained on these three original datasets have relatively good OOD performance, as shown in Table 8. We construct the biased datasets by adding the prefix "Answer: " to the human-written answers of Finance and Medicine datasets and adding the prefix "Summa-rization: " to the human-written summaries of the GovReport dataset. We do not perform any additional operations on the LLM-generated texts.

We evaluate the fine-tuning method and our method in the scenario where the source datasets

are biased datasets. As shown in Table 9, for the fine-tuning method, the OOD performance of the classification models that are fine-tuned on biased datasets is very poor, with the accuracy below 60% in most evaluated situations. The reason is that the trained classification model with supervised fine-tuning overfits dataset biases, i.e., the mapping from the prefix "Answer: " or "Summarization: " to the label "human-written". However, these dataset biases are not the universal useful features among all datasets and they can not be used for classifying the human-written and LLM-generated texts of other datasets without these biases. Thus, the OOD performance of the fine-tuning method declines much when the training dataset contains biases. Besides, as shown in Table 9, our method is robust to the dataset bias. Under all evaluated situations, the accuracy of our method is above 65%, which demonstrates that our method is still effective when the source dataset contains biases.

**Results of the Case Study.** We show examples of a human-written summary and the summary generated by ChatGPT for the same bill from the BillSum dataset (Kornilova and Eidelman, 2019). Specifically, we show the human-written summary and its ChatGPT-revised version in Figure 2. The ChatGPT-generated summary and its ChatGPT-revised version are shown in Figure 3. Some revised parts in the revised texts are highlighted in red color. From Figure 2 and Figure 3, we can see that the revisions are more obvious for the human-written summary. For example, the human-written sentence "Prescribes procedural guidelines ... for Nonperformance of Transportation" is revised into the sentence "It also establishes guidelines ... for Nonperformance of Transportation offers the service". The ChatGPT model revises the long human-written sentence into a relatively shorter sentence.

We show the similarity scores between the human-written summary and its corresponding revised text, and the similarity scores between the

| Source/Target Dataset | Finance | Medicine | Reddit Eli5 | MultiNews | GovReport | BillSum |
|---|---|---|---|---|---|---|
| | Fine-Tuning | | | | | |
| Finance | 99.66 | 87.32 | 92.61 | 75.2 | 70.75 | 82.95 |
| Medicine | 88.92 | 99.92 | 95.48 | 44.11 | 53.07 | 55.16 |
| Reddit Eli5 | 52.15 | 60.26 | 99.99 | 49.98 | 50.02 | 50.98 |
| MultiNews | 51.50 | 48.89 | 51.15 | 100.0 | 51.12 | 53.38 |
| GovReport | 75.58 | 85.81 | 72.77 | 90.19 | 96.38 | 93.74 |
| BillSum | 72.63 | 66.37 | 78.21 | 50.13 | 57.83 | 99.97 |
| | Our Method | | | | | |
| Finance | 93.72 | 88.92 | 87.60 | 77.35 | 68.26 | 78.14 |
| Medicine | 91.97 | 90.25 | 93.41 | 77.85 | 65.44 | 74.41 |
| Reddit Eli5 | 79.95 | 84.39 | 97.51 | 68.85 | 56.71 | 59.74 |
| MultiNews | 92.72 | 90.65 | 92.01 | 77.9 | 66.35 | 75.46 |
| GovReport | 92.91 | 84.95 | 81.51 | 75.18 | 69.22 | 78.79 |
| BillSum | 93.10 | 86.32 | 83.18 | 75.65 | 68.69 | 78.59 |

Table 8: Results of the fine-tuning method and our method when using original datasets as source datasets. The evaluation metric is ACC (%).

| Source/Target Dataset | Finance | Medicine | Reddit Eli5 | MultiNews | GovReport | BillSum |
|---|---|---|---|---|---|---|
| | Fine-Tuning | | | | | |
| Biased Finance | 100.0 | 53.44 | 58.55 | 53.43 | 52.57 | 56.46 |
| Biased Medicine | 59.07 | 100.0 | 72.60 | 49.78 | 50.05 | 50.05 |
| Biased GovReport | 53.44 | 54.96 | 53.17 | 57.40 | 100.0 | 52.73 |
| | Our Method | | | | | |
| Biased Finance | 94.93 | 89.65 | 88.57 | 77.85 | 67.94 | 77.79 |
| Biased Medicine | 91.76 | 92.30 | 93.64 | 77.8 | 65.02 | 74.16 |
| Biased GovReport | 93.01 | 90.37 | 90.78 | 78.03 | 88.76 | 76.49 |

Table 9: Results of the fine-tuning method and our method when using biased datasets as source datasets. The evaluation metric is ACC (%). For the experiments where the training (source) and testing (target) datasets are of the same distribution, the training (source) and testing (target) data are both with dataset biases. For the experiments of testing OOD robustness, the source dataset has the dataset bias while the target dataset (different from the source dataset) does not have the dataset bias.

| Similarity Metric | Human-Written | ChatGPT-Generated |
|---|---|---|
| BLEU Score | 0.272 | 0.794 |
| ROUGE-1 F1 Score | 0.589 | 0.904 |
| ROUGE-2 F1 Score | 0.377 | 0.823 |
| ROUGE-L F1 Score | 0.529 | 0.865 |
| BERTScore | 0.538 | 0.909 |
| BARTScore-CNN | -2.67 | -0.711 |

Table 10: Similarity scores between the human-written/ChatGPT-generated summary and its revised text in the case study, measured by different similarity metrics.

ChatGPT-generated summary and its corresponding revised text under different similarity metrics in Table 10. From the results in Table 10, we can see that the similarity score between the ChatGPT-generated summary and its corresponding revised text is higher than that between the human-written summary and its corresponding revised text no matter which similarity metric is used.

## B Details of Similarity Metrics

The details of the similarity metrics are as follows.

- BLEU score (Papineni et al., 2002): The BLEU score is calculated based on the overlap between the hypothesis's n-grams and the reference's n-grams. In our scenario, the reference is the text to be detected and the hypothesis is its corresponding revised text. The higher BLEU score indicates the higher similarity between the text and its revised text.

- ROUGE score (Lin, 2004): Similar to the BLEU score, the ROUGE score is also based on the overlap between the hypothesis's n-grams and the reference's n-grams. The higher ROUGE score indicates the higher similarity between two pieces of texts.

- BERTScore (Zhang* et al., 2020): The BERTScore is calculated based on the co-

## Human-Written Text

Authorizes passenger transportation in foreign-flag cruise vessels between Alaska ports, and between Alaska ports and those on the west coast of the contiguous States.

Prescribes procedural guidelines under which the Secretary of Transportation shall notify the owner or operator of one or more foreign-flag vessels that he shall terminate the authorization for the foreign-flag vessel to provide passenger transportation upon a showing by the owner or charterer of a U.S. cruise vessel that the U.S. vessel is offering such passenger service pursuant to a Certificate of Financial Responsibility for Indemnification of Passengers for Nonperformance of Transportation.

States that coastwise privileges granted a foreign-flag cruise vessel under this Act shall expire on the 365th day following receipt of the Secretary's termination notification.

## Revised Human-Written Text

This legislation permits travel for passengers on foreign-flag cruise vessels between Alaskan ports and between Alaskan ports and those on the west coast of the contiguous United States. It also establishes guidelines for the Secretary of Transportation to inform the owner or operator of a foreign-flag vessel that it must cease passenger transportation if a US vessel with a Certificate of Financial Responsibility for Indemnification of Passengers for Nonperformance of Transportation offers the service. These privileges for the foreign-flag vessel will expire 365 days after receiving the Secretary's notification of termination .

Figure 2: The human-written text and its ChatGPT-revised version.

## ChatGPT-Generated Text

The text discusses a congressional finding regarding the growing cruise ship industry in Alaska and the need to encourage the use of US facilities, labor, and services. It notes the importance of protecting existing US employment and economic activity and highlights the significant economic benefits received by Vancouver through the industry. The text also permits the transportation of passengers in foreign-flag cruise vessels between ports in Alaska and on the west coast of the contiguous states, provided certain conditions are met. It allows coastwise privileges for US cruise vessels and outlines a process for terminating foreign-flag vessels' service after a notification period. The text provides definitions and a disclaimer regarding the transportation of passengers and merchandise in Canadian vessels between ports in Alaska and the US.

## Revised ChatGPT-Generated Text

The text discusses a congressional finding regarding the growing cruise ship industry in Alaska and emphasizes the need to prioritize US facilities, labor, and services. It stresses the importance of protecting existing US employment and economic activity while recognizing the significant economic benefits received by Vancouver through the industry. The text also permits the transportation of passengers in foreign-flag cruise vessels between ports in Alaska and on the west coast of the contiguous states, subject to specific conditions. It grants coastwise privileges for US cruise vessels and outlines a process for terminating foreign-flag vessels' service after a notice period. Additionally, the text provides definitions and a disclaimer regarding the transportation of passengers and merchandise in Canadian vessels between ports in Alaska and the US.

Figure 3: The ChatGPT-generated text and its ChatGPT-revised version.

sine similarities of pairwise tokens' representations of two sentences. The tokens' representations are encoded by the pre-trained language model. The higher BERTScore indicates the higher similarity between two pieces of texts.

- BARTScore (Yuan et al., 2021): For the calculation of BARTScore in this paper, we use the BARTScore-CNN, which uses a BART model that is fine-tuned on the CNNDM dataset (Hermann et al., 2015). The revised text is input to the BART model and the original text to be detected is the target text. The BARTScore is the negative loss between the output logits of the BART model and the target text. The higher BARTScore indicates the higher similarity between the text to be detected and its corresponding revised text.

## C  Details of the Compared Detection Methods

The details of the detection methods we compare our method with are as follows.

(1) Log-Likelihood (Solaiman et al., 2019): The text that needs to be detected is fed into the GPT-2-medium model. The Log-Likelihood score is the negative output loss of the model. A higher Log-Likelihood score means the text is more likely to

| Statistics | Finance | Medicine | Reddit Eli5 | MultiNews | GovReport | BillSum |
|---|---|---|---|---|---|---|
| Number of Samples | 7866 | 2492 | 29190 | 4000 | 1878 | 3994 |
| Average Length of H-Texts | 175.6 | 82.2 | 147.9 | 202.1 | 303.1 | 182.1 |
| Average Length of Revised H-Texts | 136.3 | 97.6 | 102.0 | 158.8 | 223.8 | 156.3 |
| Average Length of C-Texts | 205.2 | 187.4 | 173.6 | 102.3 | 151.9 | 129.1 |
| Average Length of Revised C-Texts | 168.7 | 161.1 | 140.8 | 103.8 | 147.6 | 126.8 |

Table 11: Dataset statistics. "H-Texts" and "C-Texts" represent the human-written texts and ChatGPT-generated texts, respectively. The number of samples for each dataset takes both human-written samples and ChatGPT-generated samples into account.

be LLM-generated.

(2) Rank (Gehrmann et al., 2019): The text that needs to be detected is fed into the GPT-2-medium model. Then this method sorts the output logits in descending order for each token. After sorting, the method gets the rank of each label token. Then the method gets the final rank score by averaging the rank scores of all label tokens. A lower rank score means the text is more likely to be generated by the LLM.

(3) Log-Rank (Mitchell et al., 2023): A little different from the Rank method, the Log-Rank method just adds an additional operation by applying a log function on the rank score of each token. Similar to the rank score, a lower log-rank score means the text is more likely to be LLM-generated.

(4) Entropy (Gehrmann et al., 2019): The text that needs to be detected is fed into the GPT-2-medium model. The Entropy method calculates the entropy of the softmax logits derived from the GPT-2-medium model and then averages the entropy of each token to get the final entropy score. A lower entropy score means the text is more likely to be LLM-generated.

(5) DetectGPT (Mitchell et al., 2023): For the DetectGPT method, it perturbs the text to be detected using the T5-large (Raffel et al., 2020) model and gets the perturbed text. Then it applies a log function on the ratio of the original text's probability to the perturbed text's probability to get the final ratio. If this ratio is high, it means the text is likely to be LLM-generated.

(6) Supervised Fine-Tuning (Guo et al., 2023): For the implementation of the supervised fine-tuning method, we fine-tune the RoBERTa$_{BASE}$ (Liu et al., 2019) model on the labeled training samples to get a classification model.

## D Implementation Details

### D.1 Experimental Setting

**Datasets.** For the summarization generation task, we perform experiments on the MultiNews (Fabbri et al., 2019), GovReport (Huang et al., 2021) and BillSum (Kornilova and Eidelman, 2019) datasets. For human-written texts, we directly use the human-written summaries in the original datasets. In the main experiments, we use the representative LLM, i.e., ChatGPT (gpt-3.5-turbo), as the source model to generate the LLM-generated texts to be detected. We use the prompt "Please summarize the following text: " to make the ChatGPT model generate the summaries and get the ChatGPT-generated summaries. For the question answering task, the datasets we consider are Finance (Maia et al., 2018), Medicine (Chen et al., 2020), and Reddit Eli5 (Fan et al., 2019). In the main experiments, we use the ChatGPT-generated texts collected by Guo et al. (2023) for these three datasets. Specifically, we use the first human-written answer in the human-written answer list and the first ChatGPT-generated answer in the ChatGPT-generated answer list for each question. For some datasets, we sample some samples from the original datasets. The detailed statistics of the number of samples and the average length of samples for each dataset are shown in Table 11.

When we revise texts with ChatGPT, we use the gpt-3.5-turbo API provided by OpenAI. When generating ChatGPT-generated summaries for Multi-News, GovReport and BillSum datasets, we use the gpt-3.5-turbo API provided by OpenAI.

All experiments that call gpt-3.5-turbo API in this paper are done before June 2023, with the gpt-3.5-turbo API being gpt-3.5-turbo-0301.

**Similarity Metrics.** For the calculation of BARTScore in all experiments in this paper, we use the BARTScore-CNN, which uses a BART model that is fine-tuned on the CNNDM dataset (Her-

mann et al., 2015). For the ROUGE score and BERTScore, we use the F1 score.

### D.2 Main Experiments

**Comparisons in Zero-Shot Setting.** When calculating the AUROC performance of other zero-shot detection methods, we use the Log-Likelihood score, negative rank score, negative log-rank score, negative entropy score and log probability ratio as the prediction score, respectively, for Log-Likelihood, Rank, Log-Rank, Entropy and Detect-GPT methods. The labels of ChatGPT-generated texts are set as 1.

For the Medicine dataset, we drop a pair of human-written and ChatGPT-generated answers of one question for other zero-shot detection methods, due to the NaN value of the Log-Likelihood score. For the implementation of DetectGPT, we use the T5-large as the mask filling model and the number of perturbations is set as 10. The perturbation mode is set as "z" for DetectGPT.

**Comparisons with Fine-Tuning in Vanilla Setting.** For the experiments where the training and testing datasets are of the same distribution, we evaluate the detection performance in the classic cross-validation manner. Specifically, we first split the whole dataset into 4 parts. We train the model and test the performance for 4 turns. In each turn, we use 3 parts as the training part and the remaining 1 part as the testing part. In each turn, we use a part that is different from the testing parts in other turns as the testing part for this turn. Finally, we use the total number of correctly predicted testing samples in 4 turns to divide the total number of testing samples in 4 turns and get the final accuracy.

For the experiments of testing the OOD robustness, we fine-tune the classification model with three different seeds and get the average OOD performance as the final OOD performance for the fine-tuning method. The number of training epochs is set as 10 and the learning rate is set as $2 \times 10^{-5}$ for the fine-tuning method.

For our method, we use the BARTScore-CNN as the similarity metric to calculate similarity scores.

**Comparisons with Fine-Tuning on Biased Datasets.** For the experiments where the training and testing datasets are of the same distribution, we evaluate the detection performance in the classic cross-validation manner. Specifically, we first split the whole dataset into 4 parts. We train the

model and test the performance for 4 turns. In each turn, we use 3 parts as the training part and the remaining 1 part as the testing part. In each turn, we use a part that is different from the testing parts in other turns as the testing part for this turn. Finally, we use the total number of correctly predicted testing samples in 4 turns to divide the total number of testing samples in 4 turns and get the final accuracy.

The source dataset on which the classification model is fine-tuned is the biased dataset. For the experiments where the training (source) and testing (target) datasets are of the same distribution, the training (source) and testing (target) data are both with dataset biases. For the experiments of testing OOD robustness, the source dataset has the dataset bias while the target dataset (different from the source dataset) does not have the dataset bias.

For the experiments of testing the OOD robustness, we fine-tune the classification model with three different seeds and get the average OOD performance as the final OOD performance for the fine-tuning method.

For our method, we use the BARTScore-CNN as the similarity metric.

### D.3 Detecting Texts Generated by Various Source Models

For generating LLM-generated texts with Vicuna, we first obtain a Vicuna model using the LLaMA-7B model[2] and the delta weight[3], following the official implementation[4]. Then we use the Vicuna model to generate texts.

Some other details of the experiments are as follows. For generating LLM-generated texts with the text-davinci-002[5] and Vicuna models, we use the instruction "Please generate a long answer for the following question: ". For generating LLM-generated texts with the text-davinci-003 model[6], we do not add any additional instruction and just input the questions of the Finance and Medicine datasets into the model. For experiments of detecting the text-davinci-002 model's generated texts for the Medicine dataset, we drop 15 pairs of human-written and LLM-generated texts that correspond

---

[2]https://huggingface.co/decapoda-research/llama-7b-hf
[3]https://huggingface.co/lmsys/vicuna-7b-delta-v1.1
[4]https://github.com/lm-sys/FastChat
[5]https://openai.com/
[6]https://openai.com/

to 15 questions due to the blank answers returned by the text-davinci-002 model. We evaluate the detection performance of our method using the BERTScore and BARTScore-CNN as the similarity metric, respectively.