# OpenReview forum: "Beat LLMs at Their Own Game: Zero-Shot LLM-Generated Text Detection via Querying ChatGPT"
_EMNLP/2023/Conference — EMNLP 2023 Main_

### Official Review · Reviewer_s23n · 2023-07-26

**Soundness:** 4

**Excitement:**

3: Ambivalent: It has merits (e.g., it reports state-of-the-art results, the idea is nice), but there are key weaknesses (e.g., it describes incremental work), and it can significantly benefit from another round of revision. However, I won't object to accepting it if my co-reviewers champion it.

**Paper Topic And Main Contributions:**

The paper presents a "zero-shot black-box method"  for detecting texts that are generated by Large Language Models (LLMs).
The key idea behind their method is to revise the text to be detected using the ChatGPT model. The assumption is that if the text is  generated by an LLM, the ChatGPT model will make fewer revisions to it compared to human-written texts.

**Reasons To Accept:**

The paper presents a simple, yet effective, approach for addressing a timely task: identifying automatically generate texts and distinguishing them from human generate texts.
The approach is tested on texts acquired from six datasets pertaining to question&answering and summarization.
The authors perform experiments with texts generated by multiple source models, although they test revision only using chatgpt.

**Reasons To Reject:**

The continuous reference to appendices bothers the reading: some crucial information is reported in the appendix (such as the study on the metric threshold in app E).
The revision step of the approach employes only chatgpt, thus we do not know if the method shows similar performances also when a different model is employed for revision.

**Reproducibility:**

3: Could reproduce the results with some difficulty. The settings of parameters are underspecified or subjectively determined; the training/evaluation data are not widely available.

**Reviewer Confidence:**

3: Pretty sure, but there's a chance I missed something. Although I have a good feel for this area in general, I did not carefully check the paper's details, e.g., the math, experimental design, or novelty.

---

> ### Author Rebuttal · Authors · 2023-08-27
>
> We thank the reviewer for the valuable suggestions and insightful comments. Our point-to-point responses are shown as follows.
>
> ***Comment 1:*** **"The continuous reference to appendices bothers the reading: some crucial information is reported in the appendix (such as the study on the metric threshold in app E)."**
>
> **Response:**
> We thank the reviewer for the valuable suggestions. Following the reviewer's suggestions, we will move some crucial information that is reported in the appendix to the main body of the paper in the camera-ready version.
>
> ***Comment 2:*** **"The revision step of the approach employes only chatgpt, thus we do not know if the method shows similar performances also when a different model is employed for revision."**
>
> **Response:**
> We thank the reviewer for the valuable suggestions. Following the reviewer's suggestions, we employ some other models for revision, including the GPT-4 model and the Claude-2 model. The experimental results are shown in Table 1. From the experimental results, we can see that our method can achieve similar performance when using a different model for revision. For example, when using the GPT-4 model for revision, our method achieves an AUROC of 99.91% on the Reddit Eli5 dataset.
> | The Model for Revision | Medicine | Finance | Reddit Eli5 |
> | :-------------: | :------: | :-----: | :---------: |
> |      GPT-4      |  95.88   |  97.58  |    99.91    |
> | ChatGPT (turbo) |  96.72   |  98.73  |    99.95    |
> |    Claude-2     |  95.98   |  94.65  |    96.31    |
>
> Table 1: Performance of our method when using GPT-4, ChatGPT (turbo), and Claude-2 models for revision. The evaluation metric is AUROC (%). The similarity metric we used for our method is BARTScore-CNN. The experiments are conducted on three datasets, including Medicine, Finance and Reddit Eli5. We sample 100 human-written texts and 100 ChatGPT-generated texts for each dataset.

---

### Official Review · Reviewer_zd6D · 2023-08-03

**Soundness:** 4

**Excitement:**

3: Ambivalent: It has merits (e.g., it reports state-of-the-art results, the idea is nice), but there are key weaknesses (e.g., it describes incremental work), and it can significantly benefit from another round of revision. However, I won't object to accepting it if my co-reviewers champion it.

**Paper Topic And Main Contributions:**

This paper investigates the detection of LLM generated texts.
The auhtors propose to measure the similarity between the given text and LLM revised version, the given text would likely be LLM generated if they are similar.
They conduct extensive experiments and achieve the best average AUROC among all zero-shot methods.
When comparing with fine-tuning based models, the suggest approach shows strong OOD genelization ability.
The detection of different source LLMs (text-davinci-003/2 and vicuna) are also tested.


**Questions For The Authors:**

If human text is included in the LLM training (fine-tuning) data, can your method successfully identify it?

**Reasons To Accept:**

The idea is novel, simple and efftive. It shows good generalization.


**Reasons To Reject:**

There is no real different source model on Table 4. Vicuna is finetune on the ShareGPT data. Maybe a experiment on claude?

**Reproducibility:**

4: Could mostly reproduce the results, but there may be some variation because of sample variance or minor variations in their interpretation of the protocol or method.

**Reviewer Confidence:**

4: Quite sure. I tried to check the important points carefully. It's unlikely, though conceivable, that I missed something that should affect my ratings.

**Typos Grammar Style And Presentation Improvements:**

Perhaps it would be better to highlight some data in the table by bolding certain values, such as the maximum value of each column.

---

> ### Author Rebuttal · Authors · 2023-08-27
>
> We thank the reviewer for the valuable suggestions and insightful comments. Our point-to-point responses are shown as follows.
>
> ***Comment 1:*** **"There is no real different source model on Table 4. Vicuna is finetune on the ShareGPT data. Maybe a experiment on claude?"**
>
> **Response:**
> We thank the reviewer for the valuable suggestions. Following the reviewer's suggestions, we perform new experiments to use our method to detect texts generated by Claude-2 and Chat-bison-001 (based on PaLM2) models. We still use the GPT-3.5 turbo model for revision. From the experimental results in Table 1, we can see that our method can effectively detect texts that are generated by Chat-bison-001 and Claude-2 models. For example, our method achieves an AUROC of 98.61% when detecting texts generated by the Chat-bison-001 model on the Reddit Eli5 dataset.
>
> |     &nbsp;&nbsp;&nbsp;&nbsp;&nbsp;&nbsp;&nbsp;&nbsp;&nbsp;&nbsp;&nbsp;&nbsp;&nbsp;&nbsp;&nbsp;&nbsp;&nbsp;&nbsp;&nbsp;Source Model      | Medicine | Finance | Reddit Eli5 |
> | :-------------------: | :------: | :-----: | :---------: |
> | Chat-bison-001 (based on PaLM2) |  92.34   |  91.13  |    98.61    |
> |       Claude-2        |  85.14   |  75.13  |    95.18    |
>
> Table 1: Performance of our method when detecting texts that are generated by Claude-2 and Chat-bison-001 models. The evaluation metric is AUROC (%). The similarity metric we used for our method is BARTScore-CNN. The experiments are conducted on three datasets, including Medicine, Finance and Reddit Eli5. We sample 300 human-written texts and 300 LLM-generated texts for the Finance dataset. We sample 100 human-written texts and 100 LLM-generated texts for the Reddit Eli5 dataset.
>
> ***Comment 2:*** **"If human text is included in the LLM training (fine-tuning) data, can your method successfully identify it?"**
>
> **Response:**
> We thank the reviewer for the insightful comment. The datasets we used for evaluation are all publicly available, e.g., the Reddit Eli5 dataset. We do not know the exact training data the ChatGPT model used for fine-tuning. Some blogs and technical reports have mentioned that some Reddit data may have been used for training ChatGPT. As the ChatGPT model has been trained on massive publicly available data, some publicly available data we used for evaluating our method may be already included in the training data of the ChatGPT model.
>
> ***Comment 3:*** **"Typos Grammar Style And Presentation Improvements: Perhaps it would be better to highlight some data in the table by bolding certain values, such as the maximum value of each column."**
>
> **Response:**
> We thank the reviewer for the valuable suggestions. We will improve the grammar style and the presentation of our paper in the camera-ready version, including fixing typos and highlighting some important data in the table.

---

### Official Review · Reviewer_EYmp · 2023-08-04

**Soundness:** 3

**Excitement:**

3: Ambivalent: It has merits (e.g., it reports state-of-the-art results, the idea is nice), but there are key weaknesses (e.g., it describes incremental work), and it can significantly benefit from another round of revision. However, I won't object to accepting it if my co-reviewers champion it.

**Paper Topic And Main Contributions:**

The paper proposes a new method for detecting machine (LLM) generated text. The authors build on the core idea that machine generated text is editted fewer times in machine generated revisions of the same text. So in order to detect whether an input is machine generated, the authors query ChatGPT (GPT-3.5 Turbo) to revise the input, and use the similarity between the revised text and the original to determine whether the text is machine generated. The proposed approach beats DetectGPT and several other baselines that are mainly built on GPT-2.

**Questions For The Authors:**

Is there a reason the XSum, SQuAD, WritingPrompts datasets which are originally used to evaluate DetectGPT (your competitive baseline) are excluded in the evaluation? Including those datasets would make your method more comparable with DetectGPT. Given that DetectGPT performs worse on average than the Log-Rank baseline in your benchmarks, but outperforms it in the DetectGPT benchmarks/paper, may indicate that there is discrepancy in the set of tasks used for evaluation.

Would applying your heuristic work when the generator model and the revision model are farther away from each other? In most experiments both generated texts and revision model are GPT-3.5 turbo, with the exception of Vicuna in table 4 (which was trained using GPT-4 as well). Would the heuristic work if your texts were generated using PALM or Claude models?

Machine generated text detection is a hard problem when casted as binary classification. For instance, OpenAI recently stopped their AI generated detection classifier, citing poor performance as a reason. It seems that for any system to truly be able to detect machine generated text, much more accuracy in detection is desired. Would you consider this to be a limitation of your approach?

**Reasons To Accept:**

The proposed method brings about improvements in detecting machine generated text, which is an important task for preventing misuse of large language models. The authors provide promising findings from evaluation on several baselines across multiple datasets.

**Reasons To Reject:**

Baseline comparisons may be faulty: For the "zero shot methods for comparison" the authors use GPT-2 logits to calculate Log Likelihood, Rank, Log Rank, and Entropy for input texts. However the input generated text is sampled from GPT-3.5 turbo, a completely different model, which greatly undermines the point of using log likelihood and similar proxy measures for detecting machine generated text. The competing baseline DetectGPT uses similar zero-shot baselines for comparison, but they calculate log likelihood and other metrics using the logits from the same model that produced the generated text, making log likelihood more comparable with their method.




**Reproducibility:**

4: Could mostly reproduce the results, but there may be some variation because of sample variance or minor variations in their interpretation of the protocol or method.

**Reviewer Confidence:**

3: Pretty sure, but there's a chance I missed something. Although I have a good feel for this area in general, I did not carefully check the paper's details, e.g., the math, experimental design, or novelty.

---

> ### Author Rebuttal · Authors · 2023-08-27
>
> We thank the reviewer for the valuable suggestions and insightful comments. Our point-to-point responses are shown as follows.
>
> ***Comment 1:*** **"Baseline comparisons may be faulty: For the "zero shot methods for comparison" the authors use GPT-2 logits to calculate Log Likelihood, Rank, Log Rank, and Entropy for input texts. However the input generated text is sampled from GPT-3.5 turbo, a completely different model, which greatly undermines the point of using log likelihood and similar proxy measures for detecting machine generated text. The competing baseline DetectGPT uses similar zero-shot baselines for comparison, but they calculate log likelihood and other metrics using the logits from the same model that produced the generated text, making log likelihood more comparable with their method.''**
>
> **Response:**
> We thank the reviewer for the comment. It is well-known that ChatGPT-generated texts are of higher quality than GPT2-generated texts. As a result, people prefer to use more advanced LLMs. Thus, in our paper, we focus on detecting texts generated by advanced LLMs, e.g., ChatGPT. However, many advanced LLMs provided by commercial companies, e.g., ChatGPT, do not expose their output logits/losses at the inference time, which makes those logits/losses-based detection methods not applicable. So we use a proxy model to derive logits/losses for those detection methods (e.g., DetectGPT[1]), which rely on the access to output logits/losses of the model. Following MGTBench[2], we use the GPT-2-medium as the proxy model to derive the output logits/losses for Log-Likelihood, Rank, Log-Rank, Entropy and DetectGPT methods.
>
> References:
>
> [1] DetectGPT: Zero-Shot Machine-Generated Text Detection using Probability Curvature.
>
> [2] MGTBench: Benchmarking Machine-Generated Text Detection.
>
> ***Comment 2:*** **"Is there a reason the XSum, SQuAD, WritingPrompts datasets which are originally used to evaluate DetectGPT (your competitive baseline) are excluded in the evaluation? Including those datasets would make your method more comparable with DetectGPT. Given that DetectGPT performs worse on average than the Log-Rank baseline in your benchmarks, but outperforms it in the DetectGPT benchmarks/paper, may indicate that there is discrepancy in the set of tasks used for evaluation."**
>
> **Response:**
> We thank the reviewer for the insightful comment. Following the reviewer's suggestions, we perform new experiments to detect LLM-generated texts on a subset of the WritingPrompts dataset. Specifically, we use the GPT-3.5 turbo model to generate stories based on the prompts in the WritingPrompts dataset for deriving the LLM-generated texts. Then we use our method and other zero-shot detection methods to detect LLM-generated texts. The implementations of other zero-shot detection methods follow the paper MGTBench[1]. The experimental results are shown in Table 1. From the experimental results, we can see that the DetectGPT method performs worse than our method on the WritingPrompts dataset.
>
> |          &nbsp;&nbsp;&nbsp;&nbsp;&nbsp;&nbsp;&nbsp;&nbsp;&nbsp;&nbsp;&nbsp;&nbsp;&nbsp;&nbsp;&nbsp;&nbsp;&nbsp;&nbsp; Method         | WritingPrompts |
> | :------------------------: | :------------: |
> |       Log-Likelihood       |     99.08      |
> |            Rank            |     98.71      |
> |          Log-Rank          |     99.33      |
> |          Entropy           |     95.63      |
> |         DetectGPT          |     61.46      |
> | Our Method (BARTScore-CNN) |     99.12      |
>
>
> Table 1: Comparisons with other zero-shot detection methods on the WritingPrompts dataset. The evaluation metric is AUROC (%). We sample 2000 human-written texts and 2000 ChatGPT-generated texts for the WritingPrompts dataset.
>
> We think the discrepancy between our results and the original results reported in the DetectGPT paper is due to the **black-box setting**. DetectGPT[2] performs experiments under the **white-box** setting, where the output loss/logits of the source model are available for detection. However, we perform experiments under the **black-box** setting, where the detector cannot access the output logits/losses of the source model. Existing zero-shot detection methods, which rely on the output logits/losses of the source model, cannot be directly applied under such a black-box setting. To compare with those methods, following MGTBench[1], we use a proxy model to derive the output logits/losses for Rank, Log-Rank, Log-Likelihood, Entropy and DetectGPT methods when detecting ChatGPT-generated texts.
>
> In fact, it is normal that DetectGPT performs worse on average than the Log-Rank baseline in our benchmarks (black-box setting). The authors of DetectGPT have reported that their method is most suited to the white-box setting and the detection performance is reduced if they use a surrogate model different from the source model, which is shown in Figure 6 of the DetectGPT paper[2]. The experimental results in Table 2 of the paper MGTBench[1] have also shown that the DetectGPT method performs worse than the Log-Rank method when detecting ChatGPT-generated texts using the GPT-2-medium model as the proxy model.
>
> The black-box setting is more **practical** because public APIs or web services provided by commercial companies generally **do not** expose the output loss/logits of the model during inference, e.g., the web services of ChatGPT and Claude2. Moreover, the detector may **not know** the exact source model that generates the texts. Considering those facts, the detector cannot directly use the output logits/losses of the source model for detection in some cases.
>
> References:
>
> [1] MGTBench: Benchmarking Machine-Generated Text Detection.
>
> [2] DetectGPT: Zero-Shot Machine-Generated Text Detection using Probability Curvature.
>
> ***Comment 3:*** **"Would applying your heuristic work when the generator model and the revision model are farther away from each other? In most experiments both generated texts and revision model are GPT-3.5 turbo, with the exception of Vicuna in table 4 (which was trained using GPT-4 as well). Would the heuristic work if your texts were generated using PALM or Claude models?"**
>
> **Response:**
> We thank the reviewer for the valuable suggestions. Following the reviewer's suggestions, we perform new experiments to use our method to detect texts generated by Claude-2 and Chat-bison-001 (based on PaLM2) models. We still use the GPT-3.5 turbo model for revision. From the experimental results in Table 2, we can see that our method can effectively detect texts that are generated by Chat-bison-001 (based on PaLM2) and Claude-2 models. For example, our method achieves an AUROC of 98.61% when detecting texts generated by the Chat-bison-001 model on the Reddit Eli5 dataset.
>
> |     &nbsp;&nbsp;&nbsp;&nbsp;&nbsp;&nbsp;&nbsp;&nbsp;&nbsp;&nbsp;&nbsp;&nbsp;&nbsp;&nbsp;&nbsp;&nbsp;&nbsp;&nbsp;Source Model      | Medicine | Finance | Reddit Eli5 |
> | :-------------------: | :------: | :-----: | :---------: |
> | Chat-bison-001 (based on PaLM2) |  92.34   |  91.13  |    98.61    |
> |       Claude-2        |  85.14   |  75.13  |    95.18    |
>
> Table 2: Performance of our method when detecting texts that are generated by Claude-2 and Chat-bison-001 models. The evaluation metric is AUROC (%). The similarity metric we used for our method is BARTScore-CNN. The experiments are conducted on three datasets, including Medicine, Finance and Reddit Eli5. We sample 300 human-written texts and 300 LLM-generated texts for the Finance dataset. We sample 100 human-written texts and 100 LLM-generated texts for the Reddit Eli5 dataset.
>
> ***Comment 4:*** **"Machine generated text detection is a hard problem when casted as binary classification. For instance, OpenAI recently stopped their AI generated detection classifier, citing poor performance as a reason. It seems that for any system to truly be able to detect machine generated text, much more accuracy in detection is desired. Would you consider this to be a limitation of your approach?"**
>
> **Response:**
> We thank the reviewer for the insightful comment. We admit that machine-generated text detection is a hard problem when cast as a binary classification problem. However, formulating the problem as a binary classification task offers a straightforward way to explore the possibility of detecting LLM-generated texts. In fact, previous works [1,2] also adopt such a formulation. Moreover, our paper provides a new idea for solving the machine-generated text detection problem, i.e., the ChatGPT model makes fewer revisions to LLM-generated texts than it does to human-written texts. Our idea still holds even if the problem is not cast as a binary classification problem. Also, we use the similarity score as the likelihood that a piece of text is generated by the LLM. A higher similarity score indicates that the piece of text is more likely to be generated by the LLM. Our method is based on the statistical probability of whether a piece of text is generated by the LLM rather than just training a supervised detector.
>
> References:
>
> [1] Automatic detection of machine generated text: A critical survey.
>
> [2] DetectGPT: Zero-Shot Machine-Generated Text Detection using Probability Curvature.

---

### Meta-Review · Area_Chair_uEWy · 2023-09-18

**Recommendation:** 3

**Metareview:**

Summary:
The reviewers consider the paper to present a simple and effective approach for detecting AI generated text. The main reviewers concerns however revolve around the models used in this paper (GPT-2 and GPT-3.5 turbo) for baseline comparisons, which makes log likelihood and similar measures unreliable for detecting machine-generated text. The absence of XSum, SQuAD, and WritingPrompts datasets, originally used to evaluate DetectGPT, raises questions about the comparability of the methods. Additionally, the study's heuristic approach may not work well when the generator and revision models are significantly different, like PALM or Claude models. Lastly, the study's binary classification approach may not address the challenge of accurately detecting machine-generated text, as it can be a difficult problem requiring higher accuracy. This could be considered a limitation of the approach.

The reviewers however seem to all agree that although the paper is strong it is not so exciting. I have read the reviewers comments and the authors rebuttal comments. Two of three reviewers conclude that the paper is a sound paper with soundness score of 4 while one gives a review of 3. All reviewers on the other hand give an ambivalent score on excitement. All reviewers commented on acknowledgement of rebuttals and made appropriate changes.

Reasons to Accept:
(1) The proposed method addresses the important task of detecting machine-generated text and presents promising results on various datasets.
(2) The approach is novel, simple, and effective, offering a valuable contribution to the field.
(3) The experiments involve multiple datasets and source models, demonstrating good generalization.

Reasons to Reject:
(1) Baseline comparisons might be flawed due to the use of different source models for generating text and calculating log likelihood, making the comparison less meaningful.
(2) The reliance on appendices for crucial information and the limitation of testing revision only with ChatGPT may hinder the paper's comprehensiveness.

---

### Decision · Program_Chairs · 2023-10-07

**Decision:**

Accept-Main

**Comment:**

Summary:
The reviewers consider the paper to present a simple and effective approach for detecting AI generated text. The main reviewers concerns however revolve around the models used in this paper (GPT-2 and GPT-3.5 turbo) for baseline comparisons, which makes log likelihood and similar measures unreliable for detecting machine-generated text. The absence of XSum, SQuAD, and WritingPrompts datasets, originally used to evaluate DetectGPT, raises questions about the comparability of the methods. Additionally, the study's heuristic approach may not work well when the generator and revision models are significantly different, like PALM or Claude models. Lastly, the study's binary classification approach may not address the challenge of accurately detecting machine-generated text, as it can be a difficult problem requiring higher accuracy. This could be considered a limitation of the approach.

The reviewers however seem to all agree that although the paper is strong it is not so exciting. I have read the reviewers comments and the authors rebuttal comments. Two of three reviewers conclude that the paper is a sound paper with soundness score of 4 while one gives a review of 3. All reviewers on the other hand give an ambivalent score on excitement. All reviewers commented on acknowledgement of rebuttals and made appropriate changes.

Reasons to Accept:
(1) The proposed method addresses the important task of detecting machine-generated text and presents promising results on various datasets.
(2) The approach is novel, simple, and effective, offering a valuable contribution to the field.
(3) The experiments involve multiple datasets and source models, demonstrating good generalization.

Reasons to Reject:
(1) Baseline comparisons might be flawed due to the use of different source models for generating text and calculating log likelihood, making the comparison less meaningful.
(2) The reliance on appendices for crucial information and the limitation of testing revision only with ChatGPT may hinder the paper's comprehensiveness.